# Six Cases of Zika/Dengue Coinfection in a Brazilian Cohort, 2015–2019

**DOI:** 10.3390/v12101201

**Published:** 2020-10-21

**Authors:** Claudio Siqueira, Valéria Féres, Livia Coutinho, Isabela Junqueira, Luziane Bento, Larissa Montes, João Bosco Siqueira

**Affiliations:** 1Institute of Tropical Pathology and Public Health, Federal University of Goias, Goiania 74605-050, Brazil; luzianemb@gmail.com (L.B.); larissamontes@hotmail.com (L.M.); siqueirajb2@gmail.com (J.B.S.J.); 2Faculty of Pharmacy, Federal University of Goias, Goiania 74605-170, Brazil; vcrisrezende@gmail.com (V.F.); livia_aires@hotmail.com (L.C.); isabelacinquini@yahoo.com.br (I.J.)

**Keywords:** arboviruses, coinfection, Zika virus, dengue virus, diagnosis, signs and symptoms

## Abstract

Brazil is one of the countries which has been most affected by dengue epidemics. This scenario became more challenging with the emergence of Zika virus after 2014. The cocirculation of dengue and Zika viruses makes their diagnosis and treatment a challenge for health professionals, especially due to their similar clinical outcomes. From 2015 to 2019, we followed a cohort of 2017 participants in Goiania, Goias, Central Brazil. Febrile cases were monitored weekly, and after identification of fever, the physician performed a home visit for clinical evaluation and collection of blood/urine for diagnosis of acute dengue/Zika infection in suspected cases. Dengue acute infection was investigated by NS1 antigen and real time RT-PCR and seroconversion of anti-dengue IgM. ZIKV infection was confirmed by real time RT-PCR. Six cases of Zika/dengue coinfection among participants were reported. The clinical outcomes were suggestive for both DENV and ZIKV infection. No coinfected patient had neurological clinical manifestation, warning signs or need for hospitalization. A continuous specific laboratory confirmation for both dengue and Zika viruses should be enforced as part of the surveillance systems even in the presence of very suggestive cases of dengue fever, minimizing the risk of a late detection of ZIKV circulation.

## 1. Introduction

Brazil is one of the countries which has been most affected by dengue epidemics. This scenario became more challenging with the emergence of Zika and Chikungunya viruses after 2014, including difficulties in specific diagnosis due to cross-reaction of laboratory tests [1].

The cocirculation of dengue and Zika viruses (DENV/ZIKV) makes their diagnosis and treatment a challenge for health professionals, especially due to their similar clinical outcomes. Dengue/Zika coinfection has been described, but clinical and laboratory presentations and exact knowledge about the severity of dengue or Zika in coinfected individuals are issues that still require further investigation [2,3,4,5,6]. We present six cases of ZIKV/DENV coinfection detected in a highly endemic setting.

## 2. Materials and Methods

From 2015 to 2019, we followed a cohort of 2017 participants in Goiania, Goias, Central Brazil. Febrile cases were monitored weekly by telephone calls and by direct contact with the attending physician. After identification of fever, the physician performed a home visit for clinical evaluation and collection of blood/urine for diagnosis of acute dengue/Zika infection in suspected cases. Clinical criteria for defining suspected dengue were as follows: fever (two to seven days) and two or more manifestations: nausea/vomiting; late rash (after third day of symptom onset); myalgia/arthralgia; headache or retroorbital pain; petechiae/positive tourniquet test; leukopenia. For defining suspected Zika cases, they were early (first/second day of symptom onset) skin rash and at least one of following symptoms: low fever; conjunctival hyperemia; arthralgia; joint edema. These definitions are adopted by the Brazilian Ministry of Health [7]. The warning signs evaluated were as follows: severe abdominal pain; persistent vomiting; fluid accumulation; postural hypotension/lipothymia; lethargy/irritability; hepatomegaly; mucosal bleeding; and progressive increase in hematocrit [7].

Dengue acute infection was investigated by NS1 antigen (NS1Ag) (rapid test STRIP-Bio-Rad and Bio-Rad Platelia™) and detection of viral RNA using a one-step real-time reverse transcriptase followed by polymerase chain reaction (real time RT-PCR) as described [8], in serum collected for up to the seventh day of the onset of symptoms. The seroconversion of anti-dengue IgM/IgG antibodies was investigated using the Dengue Virus IgM/IgG Capture DxSelect™ FOCUS Diagnostics. Samples for IgM antibodies were collected for up to seven days of symptoms and between 14 and 21 days. Dengue neutralizing antibodies (NAbs) were determined in serum samples by Plaque Reduction Neutralization Test (PRNT) following a modified protocol by Morens et al. [9] and according to WHO guidelines [10]. A positive result was considered when NAbs levels were ≥1:20. PRNT reciprocal dilution positivity was defined based on a 90% reduction in plaque count (PRNT90).

ZIKV infection was confirmed by real time RT-PCR using a validated protocol [11], in serum samples collected on up to the seventh day of the onset of symptoms. Two urine samples were also collected (up to the seventh day and between the 10th and 20th day of symptoms).

Confirmed dengue cases were those with positive detection for (1) dengue NS1Ag + anti-dengue IgM; (2) DENV RT-PCR; and for Zika, those with positive detection for ZIKV RT-PCR [7].

This research was approved by the Research Ethics Committee of the Medical School of the University of São Paulo (IRB no. 507/13). Informed written consent was obtained from the parents or legal guardian of the patient before inclusion in the study.

## 3. Results

Six cases of dengue/Zika coinfection were confirmed. ZIKV RNA was detected by RT-PCR in all cases (all in urine samples); DENV RNA was detected by RT-PCR in two and NS1Ag was detected in five cases. All cases were positive for anti-dengue IgM. In cases with non-detected DENV RT-PCR, PRNT90 was performed and NAbs levels were between 1:40 and 1:1280 (DENV multitypical response). One case of negative dengue IgG prior to fever presented seroconversion (Table 1).

Table 2 presents the clinical presentation and unspecific laboratory test results. The clinical outcomes were suggestive of a ZIKV infection in five patients, with early (first/second day of symptom onset) diffuse skin rash with low or short-term fever, when present. Conjunctivitis and joint edema were also observed in two cases. Case 4 was suggestive of dengue, with high fever, headache, retroorbital pain, myalgia and diffuse maculopapular rash on the fifth day of disease. Leukopenia (leukocytes < 4000 cells/mm^3^) was found in the blood count of four patients and two patients had thrombocytopenia (platelets < 150,000 cells/mm^3^). No coinfected patient had neurological clinical manifestation, warning signs or need for hospitalization.

## 4. Discussion

Our results were different from most of the similar case studies of coinfection published so far, in which a predominance of dengue clinical outcome was observed and, although not frequent, the possibility of alarm signs and the need for intravenous hydration existed [2,3,5,6]. However, Lovine et al. [4], like us, found a clinical presentation suggestive of a Zika case, with no signs of severity warning.

No unexpected or other specific symptoms were observed. All cases had a mild presentation with general symptoms. Headache and myalgia were observed in five cases; retroorbital pain in four; nausea in three; sore throat, mild abdominal pain and diarrhea in two; arthralgia and vomiting in one case.

Leukopenia has been described for both dengue and Zika cases and thrombocytopenia is more commonly described in dengue [7]. In coinfection, the predominance of leukopenia remained in our results and in other reported cases [2,3,6]. Most reported dengue/Zika coinfection cases had normal platelet counts [5,6], as found in four in our results. However, two cases in this study presented thrombocytopenia. A similar outcome was observed in two cases reported by Dupont-Rouzeyrol et al. [2], as well as one reported by Chia et al. [3].

In this study, we confirmed diagnosis for Zika and/or dengue infections in the acute phase by direct detection of viral RNA and DENV NS1Ag in samples from symptomatic patients. Seroconversion of anti-dengue IgM antibodies was also evaluated. Silva et al. [12] tested more than 900 samples in Salvador-BA for RT-PCR DENV and ZIKV, using positivity by RT-PCR or NS1Ag as a criterion for confirming dengue cases. Several other studies have evaluated the sensitivity and specificity of NS1Ag in the diagnosis of dengue, highlighting the best sensitivity in samples tested in primary infection and specificity close to 100% [13,14,15]. These studies discussed false-positive results in some samples depending on the producer or brand of the kit evaluated. Bio-Rad Platelia NS1Ag was widely used, evaluated and stood out among the kits with the best diagnostic accuracy. After the introduction of ZIKV, new studies have discussed the cross-reactivity of NS1Ag to Zika/dengue. Gyurech et al. [16] described a false-positive DENV NS1Ag test result in a patient with an acute ZIKV infection. Estofolete et al. [17] also suggested the possibility of false-positivity for DENV NS1Ag test in one patient confirmed for acute Zika infection. This is unlike Van Meer et al. [18] who, in 39 confirmed ZIKV-infected travelers, showed that none of the ZIKV RNA positive blood samples were cross-reactive in the DENV NS1Ag ELISA. Similar results were published in a retrospective study, in which the results did not show any DENV NS1Ag cross-reactivity in 65 RT-PCR confirmed acute ZIKV samples of French Guiana [19]. In our cohort, we diagnosed 33 cases of acute ZIKV infection and none of them presented positive NS1Ag. Guidelines consider the detection of NS1Ag sufficient to confirm acute DENV infection [7]. We found, in these six cases with detected ZIKV RT-PCR, four with undetected DENV RT-PCR. All four, in addition to detected NS1Ag, were also positive for dengue IgM and presented high titers of DENV NAbs, showing the presence of infection by both viruses and not DENV NS1Ag cross-reactivity in ZIKV RT-PCR detected cases. Even so, new assessments of the relationship between acute ZIKV infection and the presence of DENV NS1Ag may still be useful to support 100% specificity of NS1 tests in ZIKV RNA positive blood samples.

## 5. Conclusions

The clinical and laboratory evolution of dengue/Zika coinfection cases is still unclear, especially regarding the severity of these cases. Our results and the other published studies so far demonstrated that a clinical diagnosis can be misleading. Cases of dengue/Zika coinfection may fulfill both the criteria of suspected dengue and Zika cases. In the absence of unique characteristics for coinfection cases, clinicians should be aware of the possibility of typical clinical presentations of both dengue and Zika, emphasizing the importance of confirmatory laboratory tests for these diseases.

We recognize that testing all suspected cases for both diseases is unfeasible, but a continuous specific laboratory confirmation for DENV and ZIKV should be enforced as part of the surveillance systems even in the presence of very suggestive cases of dengue fever, minimizing the risk of a late detection of ZIKV circulation.

ZIKV infections have become an international concern because of their potential to cause microcephaly and other neurodevelopmental abnormalities that occur as a consequence of maternal infections [20]. Early detection of ZIKV circulation is paramount in generating alerts for the population, in order to prevent new cases of congenital Zika syndrome.

## Figures and Tables

**Table 1 viruses-12-01201-t001:** Results of dengue and Zika diagnostic tests.

Specific Tests	Case 1	Case 2	Case 3	Case 4	Case 5	Case 6
Age (years)	6	9	13	17	29	40
Sex	Male	Male	Male	Male	Female	Female
DENV NS1Ag	+	+	+	−	+	+
DENV RT-PCR	−	−	−	+	+	−
DENV PRNT ^1^	+	+	+	NT ^2^	NT ^2^	+
DENV IgM	+	+	+	+	+	+
Previous DEN IgG	+	−	+	+	+	+
PI ^3^ DEN IgG	+	+	+	+	+	+
ZIKV RT-PCR ^4^	+	+	+	+	+	+

^1^ DENV PRNT90 title with values between 1:40 and 1:1280. ^2^ NT: not tested. ^3^ PI: post-infection. ^4^ Urine samples were collected on the following day after the onset of symptoms: case 1: 5; case 2: 3, case 3: 3; case 4: 5; case 5: 3; case 6: 14.

**Table 2 viruses-12-01201-t002:** Clinical and unspecific laboratory description of dengue/Zika coinfection cases in a Brazilian cohort, 2015–2019.

Clinical/Laboratory Data	Case 1	Case 2	Case 3	Case 4	Case 5	Case 6
Skin rash	+ (early)	+ (early)	+ (early)	+ (late)	+ (early)	+ (early)
Type of rash	DPMP ^1^	DPMB ^2^	DPMP ^1^	DPMP ^1^	DPMP ^1^	DPMP ^1^
Headache	+	+	+	+	−	+
Retroorbital pain	−	+	−	+	+	+
Myalgia	+	−	+	+	+	+
Conjunctivitis	+	−	+	−	−	−
Sore throat	+	+	−	−	−	−
Mild abdominal pain	+	+	−	−	−	−
Arthralgia	−	−	−	−	−	+
Joint edema	−	−	−	−	+	+
Diarrhea	−	+	−	−	+	−
Nausea	−	+	+	−	−	+
Vomiting	−	+	−	−	−	−
WS/H ^3^	−	−	−	−	−	−
Maximum tax (°C)/days	37.7/1	37.9/1	None	38.8/3	37.9/1.5	38.0/1.5
Leukocytes (cells/mm^3^)	3400	2900	4400	3000	3200	4300
Platelets (cells/mm^3^)	173,000	173,000	149,000	157,000	172,000	120,000

^1^ Diffuse pruritic maculopapular. ^2^ Diffuse pruritic morbilliform. ^3^ Warning signs/hospitalization.

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
