# Peer review of "Six Cases of Zika/Dengue Coinfection in a Brazilian Cohort, 2015–2019"

_viruses, 2020, doi:10.3390/v12101201_

Round 1

Reviewer 1 Report

This is a very important and interesting ms describing co-infections between dengue and Zika. This has been reported before, but still deserve more cases in order too be full evaluated.

I have two major issues with his ms that need to be addressed.

1 - More details in M&M. Specially why the cut-off 1:20 PRNT 90 was chosen. And why there is not Zika PRNT? Both should be done.

2 - I don't think that a positive NS1 to dengue by itself warrants dengue specificity. We have experience on this when high viral titers of Zika leads to false NS1 results for dengue. Also others like Estofolete et al (JCV) also showed false NS1 positives.

SO I do recomendado more details in patients 1-3 or a more detailed discussing arguing that they may not be real co-infection. I am still note convinced. 

Author Response

Dear reviewer, first we would like to thank you for your contributions.

Sincerely,

The authors.

Reviewer 2 Report

The diagnosis and treatment on a co-infection of dengue/Zika viruses cases is a big challenge for clinicians due to the similarities of the clinical outcomes. In this paper, Siquera et al. presented 6 dengue/Zika co-infection cases and their clinical manifestation and diagnostic test results. Their results are different from other similar coinfection case studies, in which they found that dengue/Zika coinfection could give clinical manifestation of a Zika but without severity warning. This study suggests the importance of laboratory tests for these diseases, not only relying more on clinical diagnostic.

Minor comments:

  1. On Table 1, case 4 showed NS1Ag negative, but DENV RT-PCR positive. Positive RT-PCR indicates it was on an early stage of infection (viremia), but it is usually with positive NS1Ag test. I am just wondering whats happening here, do you have speculation regarding this?
  2. On Table 1, how did you test the presence of previous DEN IgG? Did you test all participants at the begining of the surveillance or after identification of fever (by assuming IgG is not yet being produced)? The table showed that, for case 1,2,3 and 6, the DENV RT-PCR showed negative indicating it passed the viremia stage already and the amount of IgG from current dengue infection is already increasing.

Author Response

(The authors gave the same response as above.)
